# Limitations of and Alternatives to Benchmarking in Reinforcement Learning Research

## Abstract

Novel reinforcement learning algorithms, or improvements on existing ones, are commonly justified by evaluating their performance on benchmark environments and are compared to an ever-changing set of standard algorithms. However, despite numerous calls for improvements, experimental practices continue to produce misleading or unsupported claims. One reason for the ongoing substandard practices is that conducting rigorous benchmarking experiments requires substantial computational time. This work investigates the sources of increased computation costs in rigorous experiment designs. We show that conducting rigorous performance benchmarks will likely have computational costs that are often prohibitive. As a result, we question the value of performance evaluation as a primary experimentation tool and argue for using a qualitatively different experimentation paradigm that can provide more insight from less computation. Furthermore, we strongly recommend that the community switch to the new experimentation paradigm and encourage reviewers to adopt stricter standards for experiments.

## 1 Introduction

Performance evaluation has long been standard practice in reinforcement learning (RL) research. Performance, e.g., the expected cumulative reward on a particular problem, is also often the primary highlight in highly publicized works (Mnih et al., 2015; Silver et al., 2016; Vinyals et al., 2019; Berner et al., 2019; Ecoffet et al., 2021; OpenAI et al., 2019). In fact, according to our survey of NeurIPS 2022 RL papers (see Appendix A), performance evaluation is the primary form of experimentation, with 91% of empirical papers using it. This emphasis on performance evaluation has led to a standard approach for researchers: propose a new algorithm, describe it, and demonstrate its superiority to existing algorithms on benchmark problems.

The widespread use of this experiment paradigm by the research community also propagated its issues. Many articles have pointed out problems with reproducibility (Henderson et al., 2018; Islam et al., 2017; Smith, 2018; Engstrom et al., 2020) or statistical analyses (Colas et al., 2018; Agarwal et al., 2021). Other works have examined methodological issues, showing that performance evaluation is sensitive to subtle factors such as hyperparameter selection, score normalization, and the weight assigned to each task in an aggregate performance measure (Jordan et al., 2020; Whiteson et al., 2011; Balduzzi et al., 2018). These works propose new methods to control for sources of variation in performance and make the process more rigorous, including running more trials. However, these techniques are not always straightforward and can complicate the process further. With increased complexity and cost, few have adopted these more rigorous experimental practices.

Instead of attempting to improve the standard benchmarking process further, we question its value and ask, is there an alternative experimentation paradigm? Thankfully, scientists have already given us an alternative: controlled experimentation. Borrowing the terminology of Hooker (1995), we refer to *scientific testing* as the process of using carefully controlled experiments to understand an algorithm. Scientific testing has a powerful benefit over benchmarking: it can *explain why* an algorithm performs well, whereas benchmarking can only *identify that* an algorithm worked well but not why. We argue for using scientific testing as the primary means of experimentation since it provides the understanding crucial for intellectual progress.

In addition to producing deeper insights, scientific testing has another benefit; it can require fewer trials. This reduction is because a single environment is often sufficient to study the properties of an algorithm, and there is often no need to compare against many arbitrary baseline algorithms, only the ones necessary to understand the algorithm's properties. However, these benefits come at the cost of the researcher's time designing the experiments. Designing experiments that control for many factors is challenging and can require as much or more ingenuity as developing algorithms. Although experiment design seems like a drawback for scientific testing, a critical researcher will notice that designing a rigorous benchmark also requires considerable effort, as is evident by the numerous works on the topic. Moreover, any compelling experiment requires substantial effort to develop.

This work aims to answer the question: is scientific testing a viable replacement for benchmarking as the primary experimentation tool in RL? To answer this question, we provide two sources of evidence. First, we show that unless computer clusters with thousands of cores are available and it only takes a few minutes to train an agent, rigorous benchmarking will require too many trials to be practical. Second, we use exploration algorithms to demonstrate how scientific testing produces more insightful knowledge than benchmarking, using only a standard desktop computer. Based on these results, we strongly recommend that scientific testing be the primary form of experimentation and encourage reviewers to adopt stricter standards for experiments.

## 2 Background

This section provides background on RL, performance evaluation procedures, and defines notation. For simplicity of notation, we assume that an RL agent interacts with a discrete episodic Markov decision process (MDP). For each time step $t \in \{1, 2, \dots\}$, the agent observes the state $S_t \in \mathcal{S}$, takes the action $A_t \in \mathcal{A}$, receives the reward $R_t \in \mathbb{R}$, and then transitions to the next state $S_{t+1}$, where $\mathcal{S}$ is the set of all states and $\mathcal{A}$ is the set of all actions. This process repeats until, at some finite time $t$, the agent transitions to a terminal absorbing state where all rewards are zero. The initial state is sampled from the initial state distribution $d_0$, i.e., $d_0(s) \coloneqq \Pr(S_0 = s)$. The agent's objective is to find a *policy*, the process an agent uses to selects actions, that maximizes the expected discounted sum of rewards (called the *return*), i.e., $J(\pi) \coloneqq \mathbf{E}\left[\sum_{t=0}^{\infty} \gamma^t R_t\right]$, where $\gamma \in (0, 1]$ is the reward discount factor and $\pi$ is a policy.

We consider evaluation procedures that compare the performance of a set of algorithms $\mathcal{U}$ on a set of environments $\mathcal{M}$. Performance is a user-specified metric quantifying how well an algorithm $i$ solves an environment $j$, and is represented by the random variable $X_{i,j}$. The metric used in this work is $J(\pi_{\text{final}})$, where $\pi_{\text{final}}$ is the policy returned by the algorithm after a fixed number of episodes.[1] The performance evaluation procedure's purpose is to estimate an aggregate performance measure, $y_i$, for each algorithm $i \in \mathcal{U}$, using samples of $X_{i,j}$, so that $y_i$ summarizes each algorithm's performance across all environments:

$$y_i \coloneqq \sum_{j \in \mathcal{M}} \sum_{k \in \mathcal{U}} q_{j,k} \mathbf{E}\left[g_j(X_{i,j}, k)\right], \tag{1}$$

where $g_j(x, k)$ normalizes a score $x$ (a realization of $X_{i,j}$) relative to the performance of an algorithm $k$, which is taken as a baseline, and $q \in [0, 1]^{|\mathcal{M}| \times |\mathcal{U}|}$, represents a weighting for each environment and normalization baseline algorithm $k$, with $\sum_{j,k} q_{j,k} = 1$. To construct $y_i$, one needs to choose a way to sample each $X_{i,j}$, normalize an algorithm's performance on each environment, and aggregate the performance on each environment into a single score. To sample performance we use the *fully-specified* algorithms approach (Jordan et al., 2020; Patterson et al., 2020), where an algorithm is fully-specified when there are no hyperparameters the user needs to set. For the experiments below, we use the algorithms and specifications defined by Jordan et al. (2020). We review the other components of the evaluation procedure below.

For a meaningful comparison across environments, the performance metrics need to be scaled so each metric is on a similar scale for all environments. In this work, we investigate two relative normalization techniques: *performance ratios* and *performance percentiles* that scale the performance of an algorithm $i$ to

---

[1]Our conclusions will likely apply to other metrics (e.g., the average return from each episode) since the essential factor in how many trials it takes to evaluate algorithm performance is how similar the algorithms' performance metrics are for each environment and not what the metric represents.

the performance of another baseline algorithm $k$. Performance ratios are a common technique that normalizes a score $x$ proportionally to the baseline algorithm's performance, i.e., $g_j(x, k) = \frac{x - a_j}{\mathbf{E}[X_{k,j}] - a_j}$, where $a_j$ is a constant representing the minimum possible performance on environment $j$. This method assumes that the difficulty of achieving a given level of performance is linear relative to the baseline algorithm $k$'s performance. Another normalization technique, performance percentiles (Jordan et al., 2020), automatically scales the performance relative to the difficulty that $k$ had in achieving that level of performance by using the cumulative distribution function of an algorithm $k$'s performance, i.e., $g_j(x, k) = F_{X_{k,j}}(x)$, where $F_X(x) = \Pr(X \le x)$. It also has the convenient property that $\mathbf{E}[F_{X_{k,j}}(X_{i,j})] = \Pr(X_{i,j} > X_{k,j})$, which is the normalization technique proposed by Whiteson et al. (2011). For both methods, the baseline algorithm $k$ needs to be chosen. Choosing one particular baseline algorithm could unintentionally favor one particular algorithm over another in the final aggregate performance measure (Fleming & Wallace, 1986). Instead of choosing a single algorithm $k$, we use a weighted combination (defined below) of all algorithms, e.g., the weights $q$ in (1).

To obtain an aggregate measure, one needs to choose a statistical parameter to summarize the performance across environments and select a weighting for each environment. The most common statistical parameter to summarize performance across environments is the mean, but some have proposed other parameters such as the median or interquartile mean (Agarwal et al., 2021), which are robust and have greater statistical efficiency (requires fewer samples for statistical significance). However, if one cares about identifying algorithms that work reliably on each problem, then a metric that includes the lower tail of the distribution is more appropriate. Thus, to include the lower tail and for simplicity, we focus on using the mean.

Each environment does not have to be equally important according to the weights $q$ in the aggregate performance measure. One algorithm may even be ranked the best because the weighting was favorable for that algorithm. Balduzzi et al. (2018) showed that treating all environments as equally important (a uniform weighting) led to misleading claims of superhuman performance on Atari 2600 games. Instead of a uniform weighting, one can automatically determine a weighting such that no one algorithm can be ranked first solely due to the choice of weighting by using an equilibrium solution to a two player game (Balduzzi et al., 2018). Following the work of Jordan et al. (2020), we consider finding weights for both the environment and normalization baseline $k$ by finding the equilibrium solution

$$\max_p \min_q \sum_{i=1}^{|\mathcal{U}|} p_i \sum_{j=1}^{|\mathcal{M}|} \sum_{k=1}^{\mathcal{U}} q_{j,k} \, \mathbf{E}[g_j(X_{i,j}, k)],$$

where $p \in [0,1]^{|\mathcal{U}|}$ and $\sum_i p_i = 1$. The weights $q$ from the equilibrium solution can be used in (1) to get $y_i$.

Notice that due to stochasticity in the algorithms and environments, $y_i$ will usually be unknown. Thus, for the experimenter to confidently make claims about any algorithm's performance, the evaluation procedure needs to account for uncertainty by constructing confidence intervals $[Y_i^-, Y_i^+]$ for each $y_i$ such that for a confidence level $\alpha \in (0, 0.5]$,

$$\Pr(\forall i \ y_i \in [Y_i^-, Y_i^+]) \ge 1 - \alpha. \tag{2}$$

Unfortunately, applying typical concentration inequalities (Hoeffding, 1994) to construct confidence intervals as above is impossible because the aggregate measure uses weights from an equilibrium solution, which are unknown. Jordan et al. (2020) define a procedure to compute confidence intervals when using the adversarial weightings analytically, but the intervals are conservative. An alternative is to use statistical bootstrapping (Efron, 1992), which often produces narrow confidence intervals but tends to be too narrow and not provide the coverage desired in (2). While bootstrap intervals are not guaranteed to provide valid confidence intervals, they are often the narrowest and hold empirically in many settings. Since we primarily care about evaluating how much uncertainty is produced by each evaluation procedure, we use bootstrap intervals to approximate a lower bound on the number of samples needed to identify differences in algorithm performance.

## 3 Performance Evaluation

In this section, we conduct experiments to understand the impact of more rigorous performance evaluation procedures on the uncertainty in aggregate performance $y_i$. Along with the choice of normalization function,

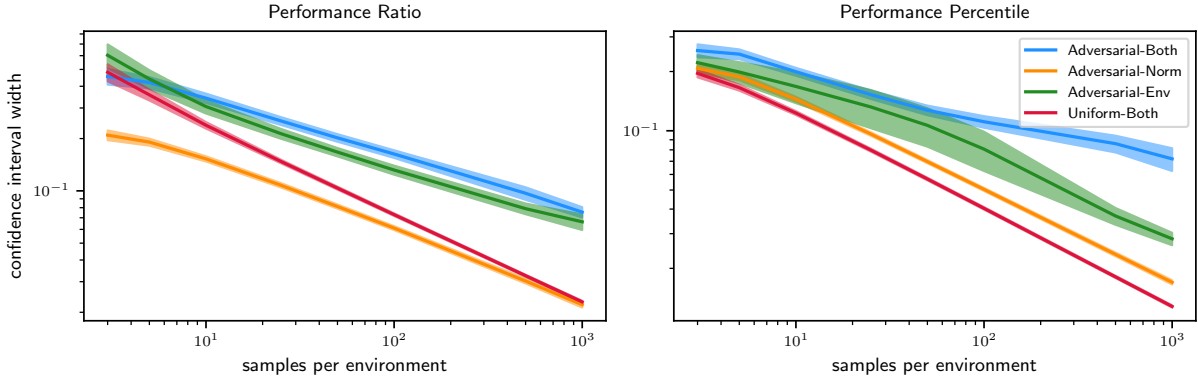

Figure 1: This plot shows the average width, over all algorithms, of the bootstrapped 95% confidence intervals versus the number of samples of each $X_{i,j}$. Different colors indicate a different aggregate weighting method. The shaded regions represent standard deviations of average confidence interval width. A total of 1,000 independent trials of the evaluation procedure were executed for each sample size. The results in the left plot use the performance ratio normalization function, while the right plot uses the performance percentile.

the experiments will investigate four variants (Adversarial-Both, Adversarial-Norm, Adversarial-Env, and Uniform-Both) of the evaluation procedure spanning the permutations of using adversarial and uniform weightings for the normalization and environment weights, e.g., Adversarial-Both use the game-based weighting for both the environments and normalization, weights whereas Adversarial-Norm uses the game-based weighting for the normalization weights and uniform weighting for the environments. Additionally, we consider variations in the number of algorithms and environments. The experiments will examine the influence of these factors on three measures of the amount of uncertainty on $y_i$: the width of the confidence intervals, the failure rate of the confidence intervals, and the number of algorithms that are not statistically significant from the best.

To measure the impact of each configuration, we will use eight standard algorithms and fourteen classic benchmark environments (listed in Appendix B). To measure the reliability of the confidence intervals, we need to have ground truth information about each algorithm's performance. Since we do not know the performance distribution for each algorithm, we will approximate it and treat the approximate distribution as the ground truth. We create the approximate distribution for each $X_{i,j}$ from approximately 334,000 executions of each algorithm-environment pair. We then create 1,000 datasets for different sample sizes, $(10, 25, 50, 100, 500, 1000)$, by sampling with replacement from the empirical distribution. We treat each dataset as a single trial of the evaluation procedure. To compute 95% confidence intervals, we use the percentile bootstrap technique with 10,000 bootstrap samples and use Boole's inequality (Boole, 1847) to correct for multiple comparisons by scaling the confidence level $\delta$ by $\delta/(|\mathcal{U}||\mathcal{M}|)$. Additionally, as a reference, we include experiments using confidence intervals leveraging $t$-distribution in Appendix C.

### 3.1 Impact of Adversarial Weighting

To understand how much each weighting method impacts the amount of uncertainty in the aggregate measure, we analyze the average width of the confidence intervals for a range of the number of samples per algorithm-environment pair; we illustrate the results in Figure 1. These plots show that adversarial weightings add significant uncertainty to the aggregate performance. For example, at 100 samples, Adversarial-Both has average confidence interval widths of 0.163 and 0.111 for the performance ratio and performance percentile normalization, respectively. In comparison, the Uniform-Both method only had average widths of 0.073 and 0.040. We also notice that performing adversarial normalization results in more uncertainty compared to a procedure that is only adversarial in environment weighting. When using performance percentiles, having adversarial environment weighting did not significantly increase the confidence interval width. However, using adversarial selection on both normalization *and* environment weightings have a combined effect on the amount of uncertainty that is greater than either part individually. This indicates that using both types of adversarial

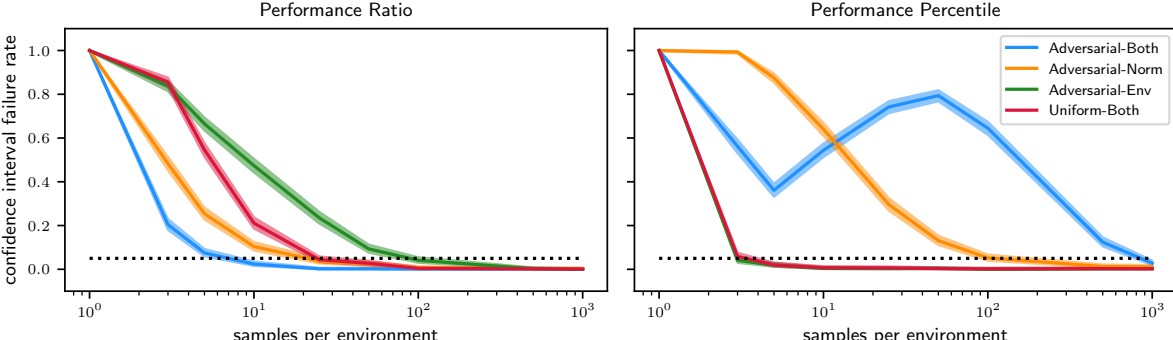

Figure 2: This plot shows the coverage probability of the bootstrapped 95% confidence intervals at each sample size. The shaded region represents 95% confidence intervals of the coverage probability using the Clopper–Pearson method (Clopper & Pearson, 1934). The dotted line indicates the target failure rate of 0.05.

weightings makes estimating the uncertainty of the aggregate performance a more difficult statistical task. Altogether, these results indicate that regardless of the normalization method, a more rigorous evaluation procedure requires many more samples to make statistically significant comparisons.

To investigate the limits of how few samples are needed to have reliable comparisons between algorithms over a set of environments, we examine the rate at which the confidence intervals fail for each method. We plot the confidence interval failure rate in Figure 2. We see that the confidence intervals for Adversarial-Both are valid at any sample size for the performance ratio. We attribute this to the large confidence intervals: even though they are valid, they cannot identify statistically significant differences, i.e., the confidence intervals overlap. The other weighting methods fail to have reliable confidence intervals at sample sizes below 100. With the performance percentile method, Uniform-Both and Adversarial-Both have high failure rates at low sample sizes. Uniform-Both can recover the desired failure rate at over 100 samples, but Adversarial-Both's failure rate remains above the target level when fewer than 1000 samples are available. Note that in the literature it is common for highly cited works to only have few samples per algorithm-environment pair, e.g., 1, 3, or 5 trials (Lillicrap et al., 2016; Bellemare et al., 2017; Hessel et al., 2018; Schulman et al., 2017; 2015; Haarnoja et al., 2018). With the high failure rate of the confidence intervals and the lack of a clear trend in Figure 2, it is unwise to blindly trust bootstrap to provide reliable confidence intervals for a small number of samples, as is currently practiced. This result further indicates that rigorous evaluation with only a few trials per algorithm-environment pair may be a rare possibility and, thus, emphasizes the difficulty of adequately comparing algorithms using standard benchmarking methods, particularly when the computational cost of running the algorithms on many environments is non-negligible.

### 3.2 Impact of Number of Algorithms and Environments

The above results may only be relevant for the specific algorithms and environments included in the evaluation. However, general properties of comparing random variables will also be present in performance evaluation. For example, it is easier to identify when two populations have different means if they are far apart than when they are close. Similarly, if one algorithm dominates others across all environments, it should be easy to identify. Additionally, we want to understand if there is a positive correlation between the number of algorithms or environments and the amount of uncertainty in aggregate performance. In this section, we conduct experiments with different combinations of algorithms and environments to study these questions.

We consider four sets of algorithms and environments to test how different algorithm and environment combinations can impact the amount of uncertainty on the aggregate performance. The four sets of algorithms contain 8, 3, 3, and 2 algorithms, respectively. One of the sets of three algorithms contains the algorithms Sarsa-Parl2, AC-Scaled, and NAC-TD, which are well separated in aggregate performance. The other set of algorithms contains Sarsa-Parl2, Q-Parl2, and AC-Parl2, the top three algorithms in aggregate

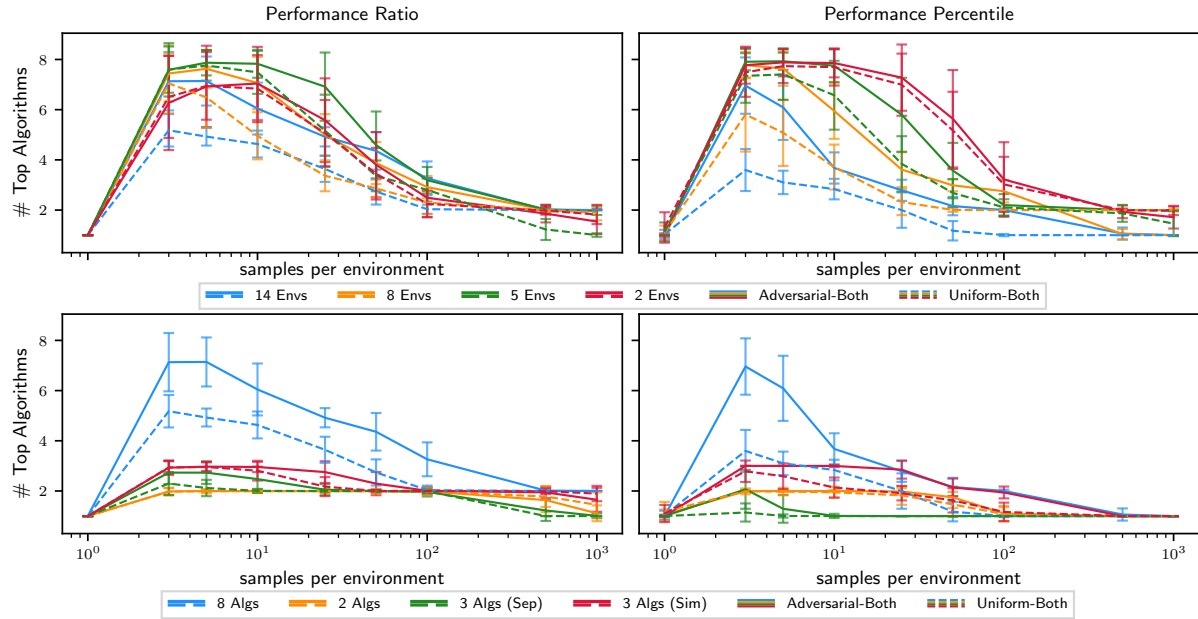

Figure 3: This plot shows the average number of algorithms that have overlapping confidence intervals with the best algorithm. The error bars represent the standard deviations. The solid lines correspond to using adversarial weightings and the dashed lines for uniform weightings. (Top) Each line color corresponds to a different group of environments denoted by the number of environments. (Bottom) Each line color corresponds to a different group of algorithms denoted by the number of algorithms. 3 (Sep) and 3 (Sim) correspond to the algorithm sets that are well separated and similar in performance, respectively.

performance. The four environment sets contain 14, 8, 5, and 2 environments, respectively. The details of each set are in Appendix B.

While the average confidence interval width allowed us to compare uncertainty in the previous section, but it does not provide a consistent measure of uncertainty when the set of algorithms or environments are changed. Instead, we define our own measure of uncertainty as the number of algorithms with overlapping confidence intervals with the first-ranked algorithm. This measure also reflects the goal of many performance evaluations where experimenters seek to identify which algorithm is the best.

We test the evaluation procedure with each algorithm and environment set and show the results in Figure 3. The exact results of any evaluation will depend on the specific combination of environments and algorithms, but there are several revealing insights. The first is evident: if algorithms have similar performance across environments, it will require more trials to identify the differences in aggregate performance. The results also suggest that unless the top algorithm is close in performance to another algorithm, reducing the number of algorithms to compare can make it easier to identify the top algorithm. In an opposite trend, the results suggest that using more environments reduces uncertainty in the aggregate performance. Furthermore, because no one algorithm dominates across all environments, these results show it can take hundreds or thousands of trials to identify the top-performing algorithm. Overall, these results show that it can be hard to predict how many trials will be necessary to identify the top-performing algorithm.

We draw a few conclusions by examining the results of this and the preceding section. First, the high failure rate of bootstrap confidence intervals in Figure 2 means that RL researchers should not rely on them for general benchmarks with less than 100 trials per algorithm-environment pair. Second, having a rigorous benchmark that uses adversarial weightings could require 1,000 trials or more to identify the top algorithm unless it dominates the other algorithms across all environments. Lastly, all of these results make it clear that benchmarking RL algorithms is an expensive experimentation practice.

Furthermore, with benchmarking requiring so many trials, obtaining reliable results can be prohibitively expensive. Consider evaluating five algorithms on ten environments, where completing a trial on each environment could take 1 minute, 30 minutes, or 4 hours. Completing even just 50 trials for each algorithm-environment pair on a desktop computer with 12 cores could take 3.5 hours, 4.34 days, or 34.7 days, respectively. These long times mean that unless researchers have access to clusters with thousands of cores, rigorous benchmarking is only feasible on very easy-to-solve problems. The standard benchmarking suites of the Arcade Learning Environment (Bellemare et al., 2013) or MoJoCo (Todorov et al., 2012) all take as least a few minutes (and often hours) to complete a single training run. Due to long compute times, researchers typically run at most five trials per algorithm-environment pair (Henderson et al., 2018; Agarwal et al., 2021).

To justify using only a handful of trials per environment, Agarwal et al. (2021) design an evaluation procedure that sacrifices rigor in the hopes of having repeatable results. In practice, their procedure does not control for hyperparameter selection, normalization bias, or environment weighting, and it uses the interquartile mean, which discards the worst 25% of runs. Furthermore, their results show that at least 20 trials are needed to compute reliable confidence intervals. Our results (Figure 2, Section 3.1) show that the reliability of the confidence intervals depends on the specific algorithms and environments being evaluated and that 20 trials is likely an optimistic estimate and insufficient to detect the best algorithms reliably.

Moreover, the complexity of the environments and the computational resources required to run algorithms is increasing. Thus, benchmarking will take even longer to complete in the future. With these insights, researchers should ask themselves, *is there any value in continuing to conduct experiments that are not rigorous and statistically reliable?*

## 4 Scientific Testing

Papers with impactful contributions go beyond presenting a new algorithm; they present new ideas that inform future algorithm design. In addition to benchmarking being computationally expensive, it only shows *which* algorithms worked well but leaves the reader to speculate *why* the algorithms performed well. Thus, there is a need for a different experimentation paradigm to serve as the primary investigative tool. *Scientific testing* sheds light on how an algorithm works and is thus a valuable tool for informing future algorithm design. In this section, we investigate scientific testing to assess if it can become the primary method of experimentation.

Scientific testing represents a broad class of experiment types, and we differentiate it from benchmarking through their respective objectives. Benchmarking seeks to learn an ordering of algorithms or identify which algorithm performs best. In contrast, scientific testing seeks to acquire or test knowledge about how a particular algorithm works. Experiments that fall under the scientific testing category may still concern performance, e.g., when testing the sensitivity of an algorithm to a specific hyperparameter. However, they may also be concerned with issues that are orthogonal to performance, e.g., identifying what kind of features neural networks learn when approximating a value function. Since the goal of scientific testing is to increase the understanding of an algorithm, it should enable others to identify and answer open questions.

Scientific testing is commonly used in RL research, but its extent varies from a single experiment that checks a specific algorithm property, such as comparing pseudo-counts to observed frames (Bellemare et al., 2016), to more comprehensive investigations that demonstrate the limitations or effectiveness of proposed methods, as in the work by Tucker et al. (2018). To determine the prevalence of scientific testing in recent RL research, we conducted a survey of NeurIPS 2022 proceedings and found that among 144 non-theory papers, 131 (91%) contained benchmarking experiments, while only 51 (35%) included scientific experiments. These results suggest that the RL community primarily relies on benchmarking, but a notable portion of works integrate scientific testing into their experiments. Our argument is for scientific testing to become the primary experimentation paradigm, which means it needs to provide sufficient knowledge for approximately 90% of papers. For more details on our survey, see Appendix A.

Below we provide examples of scientific testing to illustrate its benefits and show how it can overcome the burdens of benchmarking. For these experiments, we consider the domain of exploration algorithms, which are commonly evaluated based on their ability to solve "hard" exploration problems. Our experiments

show the inner workings of two classes of exploration algorithms: *intrinsic motivation* algorithms, which add auxiliary rewards that represent the novelty or surprise of an agent taking a particular action, and *restart-based* algorithms, which specify a particular state for an agent to start in. We describe the versions of the algorithms we investigate in the following section.

## 4.1 Exploration Algorithms

Exploration enables an agent to try new actions to identify whether they work better or worse than the other actions. A standard exploration method is to select actions randomly with some small probability. However, this randomness does not lead to the efficient discovery of good policies. A more effective family of methods is intrinsic motivation (Schmidhuber, 2010; Chentanez et al., 2004; Oudeyer et al., 2007). With intrinsic motivation, the agent receives additional rewards for taking actions that lead to surprising events or states with low visitation frequency. Over time, intrinsic rewards are adjusted based on the agent's previous actions to reflect each state's current novelty. This way, the agent keeps learning to find new unexplored areas. We study a count-based intrinsic motivation strategy (Strehl & Littman, 2008), i.e., the agent receives the reward

$$\tilde{R}_t = R_t + \frac{\beta}{\sqrt{\eta(S_t, A_t)}},$$

where $R_t$ is the external reward (comes from the environment), $\beta > 0$ controls the amount of intrinsic reward, and $\eta(s, a)$ is the total number of times the agent has taken action $a$ in state $s$.

Alternatively, instead of using reward bonuses, restart-based exploration strategies force the agent to learn how to behave from unexplored states by controlling the agent's start state distribution. With restart-based exploration strategies, the agent starts in a state sampled from a restart distribution, specified by the function $\mu \colon \mathcal{S} \to [0, 1]$, i.e., $\mu(s) \coloneqq \Pr(S_0 = s)$. The restart distribution could be a fixed distribution with high coverage of the state space, such as a uniform distribution, or it could be an adaptive distribution using a heuristic based on the agent's past experiences that determines the value of learning from a particular state (Ecoffet et al., 2021). For our experiments, we use a simplified version of the adaptive restart distribution presented by Ecoffet et al. (2021), which restarts the agent in states that are less frequently visited; the restart distribution function in particular is:

$$\mu(s) \coloneqq \frac{\eta(s)^{-1}}{\sum_{s'} \eta(s')^{-1}},$$

where, with a mild abuse of notation, $\eta(s)$ is the total number of times the agent has visited state $s$, and $\eta(s)^{-1} = 0$ if $\eta(s) = 0$. This function will sample states with a probability that is inversely proportional to how often they are visited, and will ignore states that have not been visited. Ignoring states the agent has not encountered forces the agent to learn to reach new states from the start state.

When investigating the exploration methods, we use Sarsa($\lambda$) as our base algorithm, coupled with one the methods described above. We also consider the combination of both exploration methods.

## 4.2 Scientific Testing Experiments

In this section, we present results from three experiments with exploration algorithms to highlight their behavior, as well as the usefulness of scientific testing compared to benchmarking. The first experiment studies the role of $\beta$ in influencing how much the agent explores the environment. The second experiment investigates the ability of restart-based exploration algorithms to explore hard-to-reach states. The third examines how effective each method is at learning high-performing policies across the entire state space.

As our goal is to study the fundamental properties of these algorithms, we limit our experiments to discrete MDPs. We use a variant of the four rooms MDP (Sutton et al., 1999) where there are two goal states: one ten steps away from the start state yielding a reward of 5, and one at 17 steps away from the start state yielding a reward of 10. The environment dynamics are stochastic, and with probability $\frac{1}{3}$, the state transitions as if the agent had randomly selected one of three other actions with equal probability. More details of the experiments can be found in Appendix E.

We choose a simple MDP for our study to eliminate confounding factors and accurately attribute the results to the algorithm's behavior. Our goal is also to demonstrate that even simple environments can provide valuable insights into how an algorithm works. However, scientific testing is not limited to using simple environments. We discuss implications for scaling up the results to function approximation in Section 5.

### 4.3 Intrinsic Motivation Experiments

In the first set of experiments, we test the hypothesis that increasing $\beta$, the magnitude of the reward bonus, increases the amount of entropy in the visited state-action distribution. To test this hypothesis, we run both the intrinsic motivation and the combination of restart-based and intrinsic motivation exploration methods on the four rooms domain, with $\beta \in \{0, 0.3, 0.5, 1.0, 1.5, 5.0\}$. We measure the entropy of the visited state-action pairs using an exponential moving average of the per-episode state-action counts with a weighting of $1/1{,}000$. Figure 4 shows the returns and entropies for each $\beta$, and the correlation of $\beta$ to entropy over time.

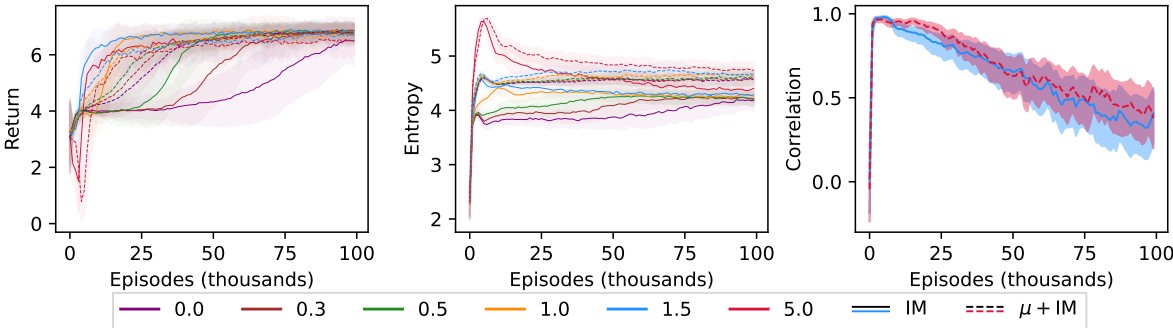

Figure 4: (Left, Middle) These plots show, respectively, the return and entropy of the visited state action distribution averaged over 100 trials. Each color represents a value of $\beta$ and the shaded regions represent the standard deviation. (Right) This plot shows the correlation coefficient between $\beta$ and the entropy of the visited state-action distribution. The colors and lines styles correspond to the algorithms Intrinsic Motivation (IM) and restart distribution with intrinsic motivation ($\mu + \text{IM}$). The shaded regions correspond to pointwise 95% confidence intervals using the Fisher transformation of the correlation coefficient (Fisher, 1921).

The results show that for both exploration methods, $\beta$ positively influences the state-action entropy, which is initially large and weakens over time. This result makes intuitive sense because the magnitude of intrinsic rewards decays over time, and the agent's behavior should converge toward an optimal policy, which is deterministic for this environment. Additionally, when $\beta$ is sufficiently high ($\beta = 5.0$), the agent practically ignores all extrinsic rewards until the intrinsic reward has been sufficiently decayed. Similarly, when $\beta$ is too low, the agent becomes "trapped" and is unlikely to explore new areas over going to the nearby goal state. This observation makes it evident that for the intrinsic motivation strategy to be effective, the $\beta$ value needs to consider the various external rewards the agent will encounter during learning. With this experiment, there is now a clear avenue for further research, e.g., discovering how to select $\beta$ so that sufficient exploration happens regardless of the value of rewards at both optimal and suboptimal policies. However, had we just benchmarked the algorithms, we would only learn whether intrinsic motivation worked, and this would only pertain to one method of choosing $\beta$, or perhaps to a single hand-tuned value of $\beta$.

### 4.4 Distance From Start State Experiments

One of the motivations for restart-based exploration methods is that they make it easier for the agent to explore states far away from the start state (Ecoffet et al., 2021). This type of motivation is a prime candidate for scientific testing! So, in our second experiment, we test the hypothesis that restart-based exploration will spend more time in states further away from the start state than intrinsic motivation algorithms. To test this hypothesis, we group the states in the far room into a set and compute the time the agent has spent in those states using an exponential moving average of state visits. To control how likely it is for the agent to enter the states in the far room by chance, we alter the random transition probability of the environment, where a

higher random transition probability increases the chances that the agent will randomly enter the far room. Additionally, we examine results using $\beta = 1.5$ and $\beta = 5.0$.

Figure 5 shows the returns and time the agent spent in the far room for each value of $\beta$. The results of this experiment show that the restart-based exploration strategy spends more time in the fourth room regardless of $\beta$ or random transition probability. This result indicates that adapting the start state distribution is likely an effective component in exploring environments where high rewarding states are far from the start state. Again, had we only considered the performance of each algorithm, we may have concluded that one method was better than another depending only on how $\beta$ was chosen and would not have learned why.

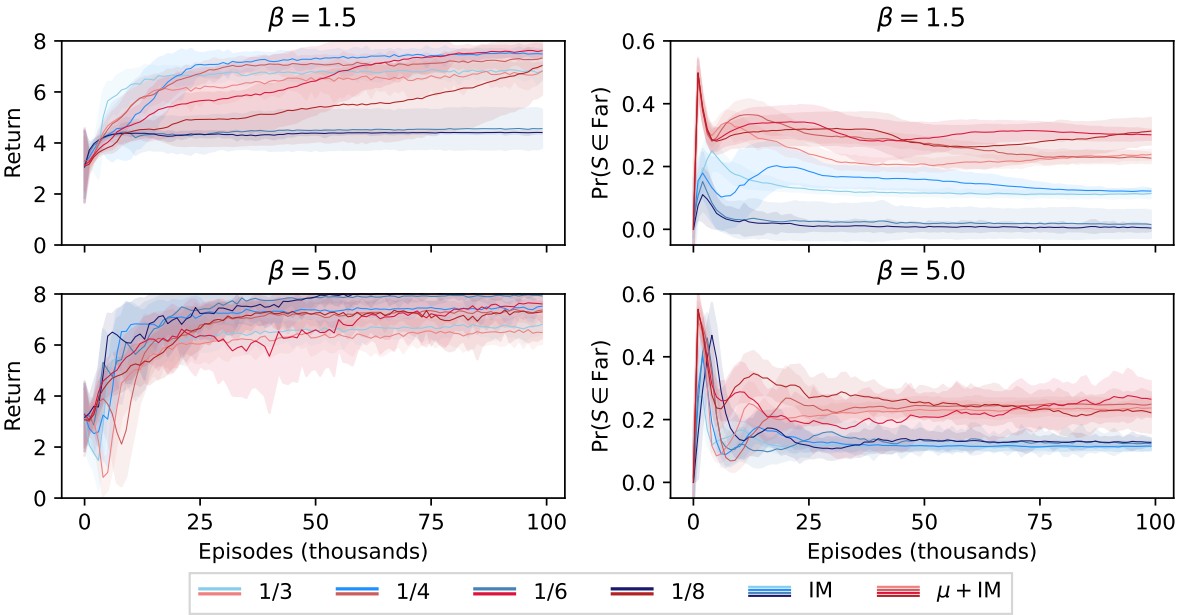

Figure 5: (Left) This plot shows the return for each algorithm for different random transition probabilities. (Right) This plot shows the proportion of time the agent was in the "Far" group of states. For all plots, each color corresponds to the algorithm Intrinsic Motivation (IM; blue lines) or restart distribution with intrinsic motivation ($\mu + $ IM; red lines). Each line style corresponds to a different random transition probability. Each line is the average of 100 trials, and the shaded areas represent standard deviations. For these plots, both algorithms use the same $\beta$, with $\beta = 1.5$ for the top row and $\beta = 5.0$ for the bottom.

## 4.5 Experiments Investigating Policy Optimization Over All States

Typically, when optimizing the return from the start state distribution, there are few opportunities to improve the policy in states that do not have a high visitation frequency. Since exploration enables the agent to visit more states, it is helpful to understand how each exploration technique can lead to policy improvement across the whole state space. Since restart distribution methods start the agent in more states, we test the following hypothesis: do exploration methods quickly improve the policy in all states, or just policy in the states frequently visited under the current policy? To test this hypothesis, we measure the distance $\|v^\pi - v^\star\|_2$ between the value function $v^\pi$ and the $\epsilon$-greedy optimal value function, $v^\star \colon \mathcal{S} \to \mathbb{R}$, where $\forall s$, $v^\star(s) = \max_{\pi \in \Pi_\epsilon} \mathbf{E}\left[\sum_{t=0}^\infty \gamma^t R_t | S_0 = s\right]$ (Sutton & Barto, 2018). Here, $\Pi_\epsilon$ is the set of all $\epsilon$-greedy policies and $\|v_1 - v_2\|_2 = \sqrt{\sum_s \left(v_1(s) - v_2(s)\right)^2}$. In addition to the exploration methods, we also evaluate the behavior of the algorithms Sarsa($\lambda$) and Sarsa($\lambda$) with the start state sampled uniformly over the state space.

Figure 6 shows the start state return and the distance to the optimal value function. The results show that the exploration methods, particularly restart distribution methods, optimize the policy over larger regions of the state space. Additionally, we see that restart distribution methods on this environment behave similarly as uniformly sampling the start state. These results suggest that controlling the start state distribution is

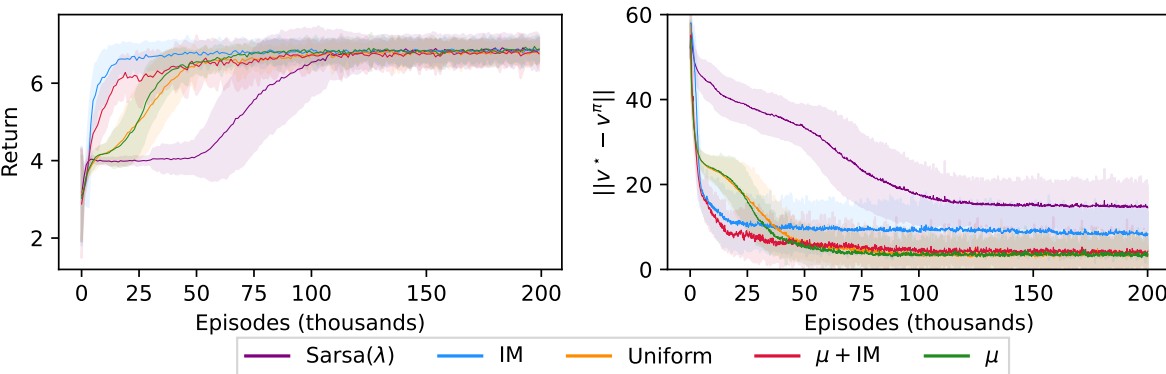

Figure 6: (Left) This plot shows the return for each algorithm over the number of episodes. (Right) this plot shows the distance of the learned policy's value function to the $\epsilon$-greedy optimal policy. Each line represents the average value computed from 100 trials, and the shaded regions correspond to the standard deviation. Each color corresponds to a different algorithm.

likely essential to learning a good policy across the entire state space. Surprisingly, none of these methods converge to a value function close to the optimal value function. As alluded to earlier, scientific testing often reveals interesting questions for further research. Here we hypothesize that this performance gap may be partly due to not having a decaying step size, and could use another experiment to test this hypothesis, but we leave that for future work.

These experiments showed four things. The first is that $\beta$ does increase the entropy of the state-action distribution, but its effect is sensitive to the reward values in the environment. The second is that restart distribution methods can spend more time away from the start state than intrinsic motivation methods. Thirdly, a restart distribution with high coverage tends to improve the policy across the whole state space. Lastly, these scientific experiments provide more information than benchmarking, using fewer computational resources. These insights can be used to design better algorithms, e.g., an intrinsic motivation algorithm with adaptive $\beta$, or identify when to use various techniques to improve performance, e.g., using a restart-based strategy in environments with hard to reach states.

### 4.6 Scientific Testings and the Computational Demands of Experiments

Using scientific testing does not guarantee that experiments will be easier to analyze statistically. However, scientific testing enables researchers to pose fundamentally different questions that benchmarking cannot answer. Although simple environments with sufficient computational resources are often useful for answering these questions, scientific testing can also be applied to more complex scenarios.

In scientific testing, researchers can specify questions that require fewer trials to test by focusing on questions where the effect size is large or the desired information that is less dependent on them. For instance, consider the results in Figure 4, where the goal is to understand how $\beta$ affects the entropy of the state-action visitation distribution. This experiment design only requires determining whether there is or isn't a correlation between $\beta$ and entropy. Regardless of whether the correlation is large or small, the result is still valuable. This example highlights a benefit of scientific testing. Even if the results do not match our expectations, they are still valuable because our goal is knowledge, not algorithm improvement.

In our experiments, we ran each algorithm one hundred times under different conditions. However, if we want to apply these experiments to larger environments, running many trials of an algorithm may become infeasible. In such cases, we need to consider asking different questions that are computationally feasible to answer. Consider the case where it is only feasible to run an algorithm once. Then benchmarking and scientific tests that examine multiple runs of an algorithm are inapplicable. Instead, the only questions we can ask about an algorithm are those that test for properties during learning. For example, instead of investigating the correlation between $\beta$ and entropy over many agent lifetimes, we can look at the influence

of $\beta$ on entropy within a single lifetime. One way to conduct this experiment is to freeze the algorithm, run it with different $\beta$ values for a fixed amount of time and reset it to its original state after each trial. This experiment design dramatically reduces the computation needed since it only requires running the algorithm with different settings for short periods. While not all questions for scientific testing can be answered in large environments, interesting ones exist for both large and small environments.

## 5  Discussion

The results in the previous section are limited to the tabular case, and it would be valuable to have experiments that show to what extent particular function approximation methods can maintain similar exploration dynamics. However, function approximation makes it more challenging to measure specific desired properties. It is tempting to ask a performance evaluation question such as "does the exploration method $X$ improve performance more across the state space than method $Y$?" We urge caution with this route as focusing only on algorithm performance will conflate issues with function approximation and algorithmic ideas. Therefore it is essential to answer questions that disentangle the two components and understand their interaction. For example, does the function approximator measure the state density well enough to provide good intrinsic rewards, or does the function approximator fail to represent specific states? Ultimately, the experiments should elucidate the algorithm's properties so others can make an informed guess about its behavior in novel problems and use insights to develop even better algorithms.

As we showed, rigorous benchmarking can be costly, which leads to the question, does it still have value? We say yes, but its role cannot be as primary evidence for proving that algorithmic ideas were correct, i.e., led to higher performance. Instead, after building up knowledge of how an algorithm works through scientific testing, benchmarks can serve as sanity or scaling checks to show that the same ideas work outside carefully controlled experiments. These check experiments can be more relaxed because their claims would be much weaker. However, it is still important to show both when an algorithm performs well and when it fails. This way, others understand the scope of the algorithm's applicability.

Finally, as a community of researchers, we are responsible for holding ourselves and each other accountable for having high standards of scientific practice. Scientific testing is one way to improve our standards and should make the researcher's life easier, in that they no longer will have to wait and hope their algorithm outperforms others. However, a change in the community will only happen if publication standards are changed to reflect our values. Based on the results in this work and the many other works analyzing experimental practices, reviewers should be confident in rejecting papers that do not conduct rigorous experiments, e.g., papers that *only* present benchmarking results, using a few trials, or that do not provide information that other researchers can use to develop better algorithms. In the end, conducting better experiments mean we will all learn more and be able to push the boundaries of knowledge quicker.

## 6  Conclusion

In this work, we provided evidence for two central claims. The first is that benchmarking can require hundreds to thousands of trials for each algorithm-environment pair to identify the top-performing algorithm in a *reliable* way, making it a burdensome task that does not have shortcuts. However, only a few papers in the RL community currently follow strict benchmarking protocols that allow for drawing meaningful conclusions. Towards improving the community's experimental practices, we show that scientific testing can be the primary experimental paradigm and gain insight into an algorithm's behavior while requiring fewer trials. We hope this work stimulates others to think about how they can create more informative experiments.

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

| Experiment Survey from NeurIPS 2022 | | | | |
| --- | --- | --- | --- | --- |
| Benchmarking | Ablation | Qualitative | Scientific | Total |
| 131 | 73 | 49 | 51 | 144 |

Table 1: This table show the number of non-theory focused RL papers containing each experiment type.

## A  Survey of RL Papers

We surveyed RL papers from the NeurIPS 2022 conference to assess the frequency of scientific testing. We searched for papers with reinforcement learning as a keyword or in the title. Our search produced 188 papers. Since theory papers contribute knowledge in the form of theorems and proofs, they should not be evaluated in the same way as empirical works. So we remove papers from the survey whose main objective is theoretical contributions. After this removal, there were 144 papers in our survey.

We looked for the presence of four types of experiments found in the paper: 1) benchmarking, experiments comparing the performance of at least two methods on one or more environments, 2) ablation studies, experiments investigating how hyperparameter choices, e.g., step sizes, network structures, algorithmic components, impact performance, 3) qualitative illustrations or examples, 4) scientific testing, experiments designed to further understanding of how an algorithm works. Table 1 shows the number of papers containing each type of experiment from this survey.

## B  Algorithm and Environment Sets

The first subset of algorithms contains Sarsa-Parl2 and AC-Parl2, which rank first and third with performance percentiles and rank second and third with performance ratio normalization (both rankings were determined with the adversarial normalization and environment weightings). The second subset is a well-separated set of algorithms Sarsa-Parl2, AC-Scaled, and NAC-TD. AC-Scaled and NAC-TD rank fifth and seventh with performance percentiles and rank first and fourth with performance ratios. The third set contains similar algorithms: Sarsa-Parl2, Q-Parl2, and AC-Parl2. Q-Parl2 ranks second and eighth with performance percentiles and performance ratios, respectively.

The 14 environments consider in the performance evaluation environment are a mixture of six continuous and eight discrete state environments. The continuous state environments are Cart-Pole (Florian, 2007), Mountain Car (Sutton & Barto, 1998), Acrobot (Sutton, 1995), and three variations of the pinball environment (Konidaris & Barto, 2009; Geramifard et al., 2015). The discrete state environments are made of two chain MDPs with ten and 50 states and deterministic transition dynamics, two Gridworlds with 5 and 10 states per side and deterministic transition dynamics. These four environments are duplicated with stochastic transition dynamics.

The first environment subset contains Acrobot, Cart-Pole, Mountain Car, Pin Ball Medium, and Pin Ball Single. The second subset includes the first plus the fifty state chain MDP, and the 10 state grid world MDPs. The third subset only has two environments, Acrobot and Pin Ball Single.

## C  Alternative Confidence Intervals Techniques

This section repeats the experiments in Section 3 but replaces the bootstrapped confidence intervals with confidence intervals that leverage the $t$-distribution. Specifically, we employ the PBP technique developed by Jordan et al. (2020) to compute confidence intervals on the aggregate performance. The PBP method works by first computing upper and lower confidence intervals on the mean normalized performance for triple of algorithm $i$, environment $j$, and baseline algorithm $k$, i.e., confidence intervals for $\mu_{i,j,k} = \mathbf{E}\left[g_j(X_{i,j}, k)\right]$. Then the minimum and maximum aggregate performance over all possible weightings that agree with these confidence intervals are computed for each algorithm $i$.

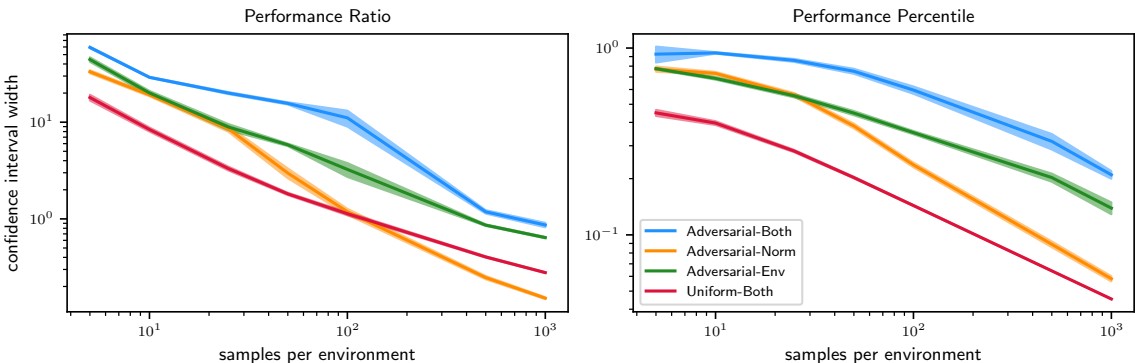

Figure 7: This plot shows the average width, over all algorithms, of the $t$-distribution-based 95% confidence intervals versus the number of samples of each $X_{i,j}$. Different colors indicate a different aggregate weighting method. The shaded regions represent standard deviations of average confidence interval width. A total of 1,000 independent trials of the evaluation procedure were executed for each sample size. The results in the left plot use the performance ratio normalization function, while the right plot uses the performance percentile.

The confidence interval technique depends on the normalization method used. For the performance ratio normalization we use the $t$-distribution to compute confidence intervals $[X_{i,j}^-, X_{i,j}^+]$ such that

$$\Pr\left(\mathbf{E}\left[X_{i,j}\right] \in \left[\bar{X}_{i,j}^-, \bar{X}_{i,j}^+\right]\right) \geq 1 - \frac{\delta}{|\mathcal{U}||\mathcal{M}|}.$$

Then lower and upper confidence intervals $[\mu_{i,j,k}^-, \mu_{i,j,k}^+]$ on the mean normalized performance are:

$$\mu_{i,j,k}^- = \frac{\bar{X}_{i,j}^- - a_j}{\bar{X}_{i,j}^+ - a_j} \qquad\qquad \mu_{i,j,k}^+ = \frac{\bar{X}_{i,j}^+ - a_j}{\bar{X}_{i,j}^- - a_j}.$$

For performance percentiles, since $\mu_{i,j,k} = \Pr(X_{i,j} > X_{k,j})$, we use Zhang-Halperin with $T$ confidence interval (Kawasaki & Miyaoka, 2010). However, due to numerical issues, this interval is not always computable, so in these cases, we resort to the DeLong $z$-interval with logit correction (DeLong et al., 1988; Perme & Manevski, 2019).

The primary thing to pay attention to in these results is that the confidence intervals are significantly wider than the bootstrap confidence intervals. This increase in width is because the uncertainty of each mean normalized performance needs to be estimated, which even increases the confidence interval width of the Uniform-Both weighting scheme. Additionally, for the methods with adversarial weightings, the optimization process to minimize (maximize) the intervals over all possible weightings creates very loose intervals. The optimization process might be improvable, but it is unlikely to produce tighter results than the bootstrap.

## D   Correcting for Multiple Comparison in Aggregate Performance

In the benchmarking procedure, we need to be able to compare the performances of all algorithms to determine a ranking of algorithms. Thus in (2), we specified that the confidence intervals for all algorithms need to have a total failure probability of at most $\delta$, i.e., $\sum_{i=1}^{|\mathcal{U}|} \delta_i \leq \delta$, where $\delta_i$ is the failure probability of the confidence intervals for algorithm $i$. There are many ways to choose $\delta_i$, but a common one is to scale $\delta_i$ inversely by the number of confidence intervals being compared, i.e., $\delta_i = \delta/|\mathcal{U}|$. However, following the work of Jordan et al. (2020), we noticed that only scaling by $|\mathcal{U}|$ led to higher failure rates of the confidence intervals, particularly at low sample sizes. So in our experiments, we used a scaling of $\delta_i = \delta/(|\mathcal{U}||\mathcal{M}|)$. In Figure 10, we show the confidence interval failure using the $|\mathcal{U}|$ scaling.

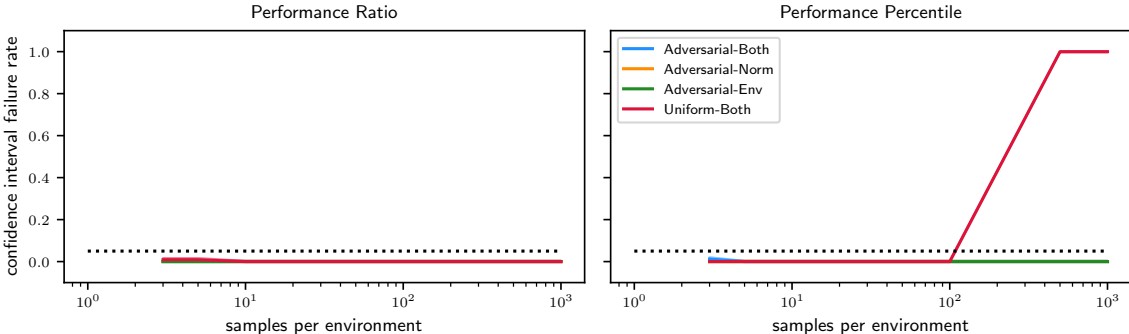

Figure 8: This plot shows the coverage probability of the $t$-distribution-based 95% confidence intervals at each sample size. The shaded region represents 95% confidence intervals of the coverage probability using the Clopper–Pearson method (Clopper & Pearson, 1934). The dotted line indicates the target failure rate of 0.05.

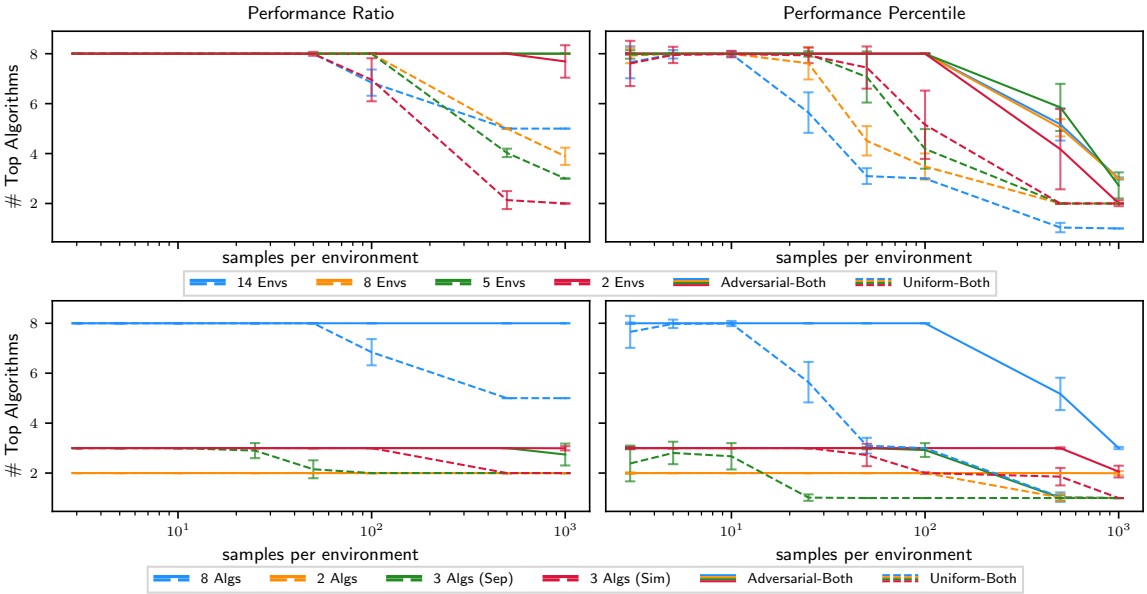

Figure 9: This plot shows the average number of algorithms that have overlapping confidence intervals with the best algorithm. The error bars represent the standard deviations. The solid lines correspond to using adversarial weightings and the dashed lines for uniform weightings. (Top) Each line color corresponds to a different group of environments denoted by the number of environments. (Bottom) Each line color corresponds to a different group of algorithms denoted by the number of algorithms. 3 (Sep) and 3 (Sim) correspond to the algorithm sets that are well separated and similar in performance, respectively.

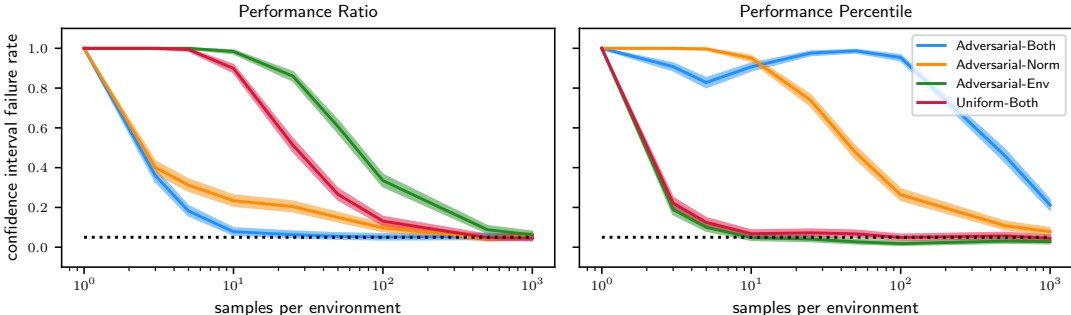

Figure 10: This plot shows the coverage probability of the bootstrapped 95% confidence intervals using the confidence level $\delta/|\mathcal{U}|$ for each algorithm's confidence interval. Solid lines show the average failure rate of the confidence intervals from 1,000 samples. Shaded regions correspond to point-wise 95% confidence intervals.

## E   Scientific Experiment Details

We had to make several design decisions when using the algorithms to run the scientific testing experiments. One of the most important is the step size. We set the step size to be $\eta = p/4$, where $p$ is the probability of random transition occurring. Intuitively, this step size allows the algorithm to average the results over a period of time that, in expectation, one random transition will occur. The division by 4 helps account for the number of actions. We found that this step size scheme made it so we did not need to tune the step size for each change in $p$. The other hyperparameters we chose were $\epsilon = 0.02$ and $\lambda = 0.9$. Changing these hyperparameters will impact the experiment results, particularly $\epsilon$, which controls how much randomness the policy has. However, the primary effects we are trying to access are independent of these parameters.

