# OpenReview forum: "Limitations of and Alternatives to Benchmarking in Reinforcement Learning Research"
_TMLR — Rejected by TMLR_

### Review · Reviewer_Ax2s · 2023-02-23

**Summary Of Contributions:**

The authors argue for a different way to evaluate RL algorithms, through experimentation rather than benchmarking. They propose to incorporate weighting over environments and scaling of reward and look at how incorporating these affect the uncertainty estimates of the results, and therefore statistical significance, in comparison to the standard, uniform weighting we use in benchmarking.

They find that, regardless of normalization method, rigorous evaluation requires many samples to make statistically significant comparisons. As a somewhat negative result, they show that we cannot obtain reliable results with less than 100 samples (compared to the standard use case of 10 seeds).

In light of this negative result, the authors argue for scientific testing to understand why algorithms perform well. As an example of this, they focus on a set of exploration strategies and evaluate them in a simple, discrete MDP in terms of metrics like entropy, distance from start state, and policy optimization.

**Audience:**

No

**Claims And Evidence:**

No

**Requested Changes:**

I think for this paper to have true impact there need to be real, actionable suggestions that are not yet widely adopted by the RL community. I think what is currently here in terms of scientific testing is already widely adopted.

**Strengths And Weaknesses:**

Strengths: The authors are highlighting an important issue, namely that to measure uncertainty and compare performance across baselines with statistical significance can be computationally prohibitive. They argue that scientific testing has a place in RL and should be emphasized.

Weaknesses: I don't think anyone in the RL research community disagrees that scientific testing is also important and necessary in a paper. I believe that most good papers already do scientific testing - through ablations, experiments in toy settings where we can compare to ground truth or isolate performance differences due to algorithmic components. I also don't think anyone would seriously accept that both these components aren't necessary - without scaling algorithms to larger and more complex benchmarks, we can't actually determine how well these algorithms will work in practice and in large-scale environments. At some point, these large scale environments are needed for us to even understand what problems are still unsolved and are interesting open areas for additional research.

---

> ### Author Response · Authors · 2023-03-15
> **Response to your review**
>
> Thank you for your review. In light of your comments, we have conducted a survey to show that scientific testing has yet to be widely adopted. Please see our updated paper for more information.

---

### Review · Reviewer_SL9E · 2023-02-27

**Summary Of Contributions:**

The paper argues that rigorously benchmarking performance is computationally expensive. To mitigate this, the paper makes a case for discarding benchmarking in favor of scientific testing for understanding how an algorithm works, which can provide insights while being less computationally expensive (by using simpler tasks as opposed to complex tasks).

**Audience:**

Yes

**Claims And Evidence:**

No

**Requested Changes:**

Comments / Questions / Requested changes:

> .. Standard formula for researchers to follow: design a new algorithm, describe it, and experiment.

- Why do you think this is the status quo? May be worthwhile to add to the paper before arguing for replacing it?

> This reduction is because a single environment is often sufficient to study the properties of an algorithm.

- Scientific testing on a single environment can be quite limited -- for example, an algorithm can work well on CartPole but falls on its face on other environments. This is somewhat analogous to how benchmarking on a single environment is limited.

> Although experiment design seems like a drawback for scientific testing, a critical researcher will notice that designing a rigorous benchmark also requires considerable effort.

- It is worthwhile to note that designing a rigorous benchmark has an one time cost,  which everyone might benefit from,  while scientific testing requires considerable experiment design and effort each time.

> “For identifying algorithms that work reliably on each problem .. We include the lower tail of the distribution and for simplicity, we focus on using the mean.

- This is a very limiting assumption for diverse environments with unbounded scores: e.g., normalized mean scores on the prevalent Atari benchmark would often prefer algorithms that perform really well on a couple of outlier tasks.  When benchmarking, if we really care about the lower tail of the distribution, the optimality gap (that is, clipped mean) from the statistical precipice / Agent57 paper or some other robust metric is much better to use. [Request] As such, I’d like the authors to run the experiments with one such metric. If mean is still preferable, the paper should clearly mention this as a key limitation of this work.

> Use of adversarial weighting in Section 3 on Performance evaluation seems redundant.

- To the best of my knowledge, the work of Jordan et al. (2020) on adversarial weighting is not adopted much by the RL community.   As such, showing its limitations (in terms of larger uncertainty) doesn't add much for arguing against rigorous benchmarking in RL. I’d suggest that maybe this can go in the appendix with focus on results on the uniform weighting, the prevalent practice.

> Bonferroni correction seems to be over-conservative.

- Can you elaborate why the `|U||M|` Bonferroni correction is needed? Assuming we are interested in comparing a given algorithm to all other algorithms, the correction should simply be `|U|`.  Intuitively, different envs possibly capture different characteristic of an algorithm and given that performance is normalized, they should all be counted as additional runs which decrease our uncertainty (and thereby increasing our confidence). For example, if we use |M| copies of the same environment -- this should not result in decreasing our confidence.

> Instead, we define our own measure of uncertainty as the number of algorithms with overlapping confidence intervals with the first-ranked algorithm.

- Overlap in confidence intervals doesn't imply lack of significance – two algorithms can be overlapping but one may still statistically significantly outperform the other. As such, the measure seems a little crude and maybe it is worthwhile mentioning the above caveat.

> Papers with impactful contributions go beyond presenting a new algorithm; they present new ideas that inform future algorithm design.

- This is typically true for good RL papers that use benchmarks -- they present an idea for which benchmarking presents additional supporting empirical evidence.

> Benchmarking only shows which algorithms worked well but leaves the reader to speculate why the algorithms performed well.

- Ablation studies, which is often needed for empirical papers that use benchmarks, tell us what components contribute to an algorithm’s performance.


> As our goal is to study the fundamental properties of these algorithms, we limit our experiments to discrete [four-room] MDPs.

- Would these conclusions generalize to more complex problems? Use of simple environments also limits scientific testing – as such the computational cost of testing would also increase as we might want to focus on more complex problems.


> In the first set of experiments, we test the hypothesis that increasing β, the magnitude of the reward bonus, increases the amount of entropy in the visited state-action distribution.

- This seems to be well thought-out but there are way too many uninteresting or obvious hypotheses which can be tested but may not add much value. Q: Would scientific testing burden the reviewers with this identification while creating a deluge of papers?

> Reviewers should be confident in rejecting papers that do not conduct rigorous experiments, e.g., papers that only present benchmarking results, using a few trials.

- By this logic, a lot of influential follow-ups to DQN (e.g., double Q-learning, dueling architectures), which only present benchmarking results using a few trials, would have been rejected. The field would have been much worse without such contributions.

Minor comments:

- Performance normalized relative to the performance of an algorithm k.
Nit: algorithm k might be more broad to include humans too. This will capture the widely used human-normalization scheme.

- We examine the rate at which the confidence intervals fail for each method. Please clarify how this failure rate is calculated?

- Nit: Furthermore, normalization in (1) is dependent on set of algorithms being used has the downside that ranking depends on that particular set -- this makes it harder to compare results across papers.

- Related work: IIRC Agarwal et. al (2021) presented IQM as an alternative to median – they also propose performance profiles and other metrics like probability of improvement for reliable benchmarking practices.  Other related works should also be discussed in more detail.


**Strengths And Weaknesses:**


Strengths
- The scientific testing experiments focusing on exploration in Section 4 are nicely done.
- The focus of this paper is on a timely and important topic.
- The paper argues for scientific testing, which sheds light on how an algorithm works, and is valuable for designing better algorithms.

Weaknesses
- I feel scientific testing can nicely complement benchmarking as opposed to replacing it. However, the paper argues for the latter without presenting any strong arguments other than benchmarking being costly.

- Given the aim in ML research to tackle more complex problems over time, the computational cost of running experiments is going to increase regardless. Instead, the focus would be on developing algorithms that can be computationally efficient on such problems as opposed to running away from complex problems – Atari 100k, despite its benchmarking problems, shows a nice example: We now have agents (e.g., Efficient Zero) that can learn to play Atari games in 2 hrs of game play!

- When talking about computational cost, the paper often conflates use of complex tasks with benchmarking. Benchmarking / Scientific testing can also use simpler environments, but their conclusions may not generalize to more complex environments.

- Key assumptions (like use of mean, adversarial weighting etc.) do not capture the current practices for RL benchmarking and limit the scope of contributions. Some technical aspects (e.g., Bonferroni corrections) are not clear and / or seem redundant.

- We already struggle from having a deluge of RL algorithms that do not work well beyond toy problems and simply getting rid of benchmarking on complex tasks would likely make the situation worse.

- The paper doesn’t do a good job of mentioning the limitations / issues with scientific testing – for one, why is this approach not widely adopted?

---

> ### Author Response · Authors · 2023-03-15
> **Response to your review part 1**
>
> Thank you for your detailed review! We have answered your questions and responded to some of your comments below. We want to point out that our experiments on a discrete MDP do not imply that scientific testing can only be used with toy environments. It is the opposite. Scientific testing will allow us to answer questions that benchmarking cannot and continue to do so as the complexity of environments increases. We have made a note of this in section 4.6.
>
>
>
> >“Can you elaborate why the |U||M| Bonferroni correction is needed? Assuming we are interested in comparing a given algorithm to all other algorithms, the correction should simply be |U|”
>
> Thanks for pointing this out. This correction was performed by Jordan et al. (2020), and we also found that scaling of |U| led to higher failure rates of the confidence intervals. We added these results to Appendix D. Also, note that with adversarial environment weightings, there is a comparison of |U||M| random variables in the computation of the aggregate performance.
>
>
> >“Please clarify how this failure rate is calculated?”
>
> We know the actual aggregate performance for each algorithm because we treat a large sample size of performance for each algorithm as a ground truth distribution and create i.i.d. samples from these distributions. So using the entire data set, we can compute the actual aggregate performance of each algorithm. Then we find the proportion of times over all trials that the confidence interval of any algorithm does not contain the mean. This proportion of times failing to capture the actual aggregate performance is the empirical failure rate of the confidence interval.
>
>
> >“Would these conclusions generalize to more complex problems? Use of simple environments also limits scientific testing – as such the computational cost of testing would also increase as we might want to focus on more complex problems.”
>
> This environment is meant to serve as an example of some of the ways to do scientific testing. We discussed the limitations of this in Section 5. Also, note that we are not arguing for only using trivial toy problems to conduct scientific tests. Scientific testing can be used with any environment.
>
>
> >“Q: Would scientific testing burden the reviewers with this identification while creating a deluge of papers?”
>
> This argument could be applied to benchmarking, i.e., researchers can create many uninteresting benchmarks. Figuring out how to pose interesting questions is the process of learning to become a good scientist.
>
>
>
> >“Given the aim in ML research to tackle more complex problems over time, the computational cost of running experiments is going to increase regardless. Instead, the focus would be on developing algorithms that can be computationally efficient on such problems as opposed to running away from complex problems – Atari 100k, despite its benchmarking problems, shows a nice example: We now have agents (e.g., Efficient Zero) that can learn to play Atari games in 2 hrs of game play!”
>
> Can the reviewer clarify their argument here? The logic of the argument feels inconsistent. If environments increase in complexity, algorithms will take longer to solve them, no matter how good we make the algorithms. There will always be a limit on how quickly a problem can be solved. So benchmarking times will necessarily increase if we try to benchmark on more and more complex problems over time. This line of reasoning should lead to the question: what kind of experiments can one run if one can only train the agent once? Rigorous benchmarking is not possible in this setting due to stochasticity. In this case, the only avenue that may be left is scientific testing, where researchers can test for the properties of an algorithm during a single trial.
>
>
>
> >“Scientific testing on a single environment can be quite limited -- for example, an algorithm can work well on CartPole but falls on its face on other environments. This is somewhat analogous to how benchmarking on a single environment is limited.”
>
> True, experiments can always be designed so that their properties do not generalize beyond the specific test setting. These are poor experiments. Designing good experiments to illustrate knowledge that generalizes is part of what makes research challenging and fun to read. As a question to the reviewer, what information do you think, in general, will be transferable to other settings, measuring how well an algorithm works under one specific measure of performance or experiments that illuminate how an algorithm works? This paper argues for the latter. In agreement with the reviewer's point that if one does not design sets of experiments that generalize to other settings, scientific testing will also not provide helpful predictions. So the key, as always, is designing useful and feasible sets of experiments :)

---

> ### Author Response · Authors · 2023-03-15
> **Response to your review part 2**
>
> >“[Request] As such, I’d like the authors to run the experiments with one such metric. If mean is still preferable, the paper should clearly mention this as a key limitation of this work.”
>
> Notice that the mean of performance percentiles is the probability of improvement, one of the robust metrics you mention.
>
> >“It is worthwhile to note that designing a rigorous benchmark has an one time cost, which everyone might benefit from, while scientific testing requires considerable experiment design and effort each time.”
>
> This argument assumes that everyone wants to measure performance in the same way. The reality is that even in benchmarking, researchers need to design a benchmark to answer the specific hypothesis (or research question) they posed. Because there will be variance in these questions, there will not be a single benchmarking procedure everyone can follow.
>
>
> >“Overlap in confidence intervals doesn't imply lack of significance – two algorithms can be overlapping but one may still statistically significantly outperform the other. As such, the measure seems a little crude and maybe it is worthwhile mentioning the above caveat.”
>
> I think there is confusion between making claims on p-values and statistical significance. From Wikipedia: "In statistical hypothesis testing,[1][2] a result has statistical significance when a result at least as "extreme" would be very infrequent if the null hypothesis were true." There are two primary ways of checking for significance, p-values being below the significance level and confidence intervals not overlapping with some target value (or another confidence interval in our case). These processes will be the same when the distribution used to compute the p-value is the same as the one used to compute the confidence intervals, e.g., Student's t-test and confidence intervals based on Student's t-distribution. In our case, we use confidence intervals to detect statistical significance.
> For example, we want to compare the aggregate performance of $y_i$ and $y_j$ to see if $y_i > y_j$. If we have confidence intervals $Y_i^-$ and $Y_j^+$ such that $\Pr(y_i \ge Y_i^-) \ge 1- \alpha/2$ and $\Pr(y_j \le Y_j^+) \ge 1- \alpha/2$. Then if $Y_i^- > Y_j+$, we can say that $y_i > y_j$ with confidence level $\alpha$, i.e., there is a statistically significant difference. Conversely,  if the confidence intervals overlap, i.e., $Y_i^- \le Y_j^+$, then we cannot say that $y_i > y_j$ with high confidence, i.e., there is no statistically significant difference.
>
>
> >“Ablation studies, which is often needed for empirical papers that use benchmarks, tell us what components contribute to an algorithm’s performance.”
>
> Yes, ablation studies can illustrate which components contribute to performance, which can provide valuable information. However, they do not reveal why the algorithm worked well. Consider the case C51 algorithm and the paper by Bellemare et al. (2017). In this paper, they conducted an ablation varying the number of atoms used to represent the distribution of the value function. This study found that a more expressive distribution led to high performance. However, later works (Lyle et al. 2019, Dabney et al. 2021) showed that the only benefits of a distributional value function were that it helped representation learning, not decision-making. These later works used scientific testing to understand how the algorithm works.
>
> Bellemare, Marc G., Will Dabney, and Rémi Munos. "A distributional perspective on reinforcement learning." In International Conference on Machine Learning, pp. 449-458. PMLR, 2017.
>
> Lyle, Clare, Marc G. Bellemare, and Pablo Samuel Castro. "A comparative analysis of expected and distributional reinforcement learning." In Proceedings of the AAAI Conference on Artificial Intelligence, vol. 33, no. 01, pp. 4504-4511. 2019.
>
> Dabney, Will, André Barreto, Mark Rowland, Robert Dadashi, John Quan, Marc G. Bellemare, and David Silver. "The value-improvement path: Towards better representations for reinforcement learning." In Proceedings of the AAAI Conference on Artificial Intelligence, vol. 35, no. 8, pp. 7160-7168. 2021.
>
>
> >“To the best of my knowledge, the work of Jordan et al. (2020) on adversarial weighting is not adopted much by the RL community.”
>
> Note that the idea of adversarial weighting comes from Balduzi et al. (2018) to combat favoring one algorithm due solely to a weighting of the environments. We want to show that as benchmarks try to overcome this issue, then it makes benchmarking more expensive. As you point out, most of the community does not try to solve this issue. Still, it is important to understand the cost if people were to try and address the weighting issue. We use Jordan et al.'s game formulation for computational efficiency and because their code was open source.

---

> ### Author Response · Authors · 2023-03-15
> **Response to your review part 3**
>
> >“This is a very limiting assumption for diverse environments with unbounded scores: e.g., normalized mean scores on the prevalent Atari benchmark would often prefer algorithms that perform really well on a couple of outlier tasks.”
>
> This argument is only valid under particular conditions. If performance ratios are used with a poor-performing algorithm in the denominator and that high-performing algorithm is not proportionally outperformed by another algorithm on another environment. Using an arithmetic mean over performance ratios is one of the issues pointed out by Fleming and Wallace (1986). This issue is resolved by considering different normalization techniques, using adversarial normalization, or using adversarial environment weightings. In the adversarial normalization case, an algorithm’s score is at most one because the best algorithm will receive the highest weighting in the denominator. Similarly, in the adversarial environment weighting case, one algorithm cannot be the best solely by exploiting an unusually high-performance metric on several environments. These issues are easy to create in a benchmark, which is why one needs to be very careful in how they design their benchmarking procedure to reflect the properties of algorithm performance they care about.

---

### Review · Reviewer_znDc · 2023-03-02

**Summary Of Contributions:**

This paper provides empirical statistical evidence of the difficulty of performing rigorous performance evaluation of RL algorithms. It points out that performing sufficient experiments to compare algorithms’ expected return, and specifically to construct reliable confidence intervals over expected returns, can take hundreds or thousands of experiments.

The paper goes on to recommend an alternative method of evaluating RL algorithms, which they denote *scientific testing*. This refers to investigations into how an algorithm works; for example, evaluating how often an exploration algorithm visits states far from the start state.

**Audience:**

Yes

**Broader Impact Concerns:**

None.

**Claims And Evidence:**

Yes

**Requested Changes:**

It would be very helpful to disentangle discussion of *policy performance* from *algorithm performance*, given that algorithms are never evaluated with i.i.d. samples of (one training run, one evaluation episode).

There needs to be some acknowledgement of the prior existing of scientific testing.

There should be some discussion of why the kinds of metrics called scientific testing are easier to compute with confidence than expected return, if they are, and why that no longer matters, if they are not.

Some of the charts are very difficult to read. It is not easy to tell the difference between the dashed, dotted, and solid lines on the plots, and the legends for two of the dashings are indistinguishable (I think it was meant to be solid and long dash?). Confidence intervals are often shown even when they overlap to the point that I can’t tell which CI corresponds to which line / method.

In Figure 5 the two rows of figures should be labeled, as it is not easy to tell them apart.

**Strengths And Weaknesses:**

### Strengths

The authors describe a real question in the RL community: how confident can we be in reported scores? What evidence is required to conclude that one algorithm is better than another?

The conclusions about the number of experiments required to trust confidence intervals was enlightening. The evaluation of the impact of adversarial weighting on the reliability of confidence intervals was especially interesting — it would make me hesitate to use adversarial weighting.

The writing is clear and to the point throughout.

The experiments in the scientific testing section, which are largely meant to serve as an example, are actually quite nice on their own, and I particularly enjoyed the impact of changing the start state distribution on the global optimality of the policy.

### Weaknesses

**Multiple axes of algorithmic evaluation**

Typically in RL evaluation there are several axes of aggregation that can be done, and each of them have quite different behavior. This paper mixes them together, leading to analysis that is not entirely actionable. Concretely, three axes:

1. There is stochasticity in the estimation of the return for a specific policy. That is, given a fixed policy, each episode has a random return.
2. There is stochasticity in the training trajectory for an algorithm. Starting with random seed 0 instead of 1 may lead to a policy with a very different *true* expected return.
3. There are multiple environments to be used for evaluation.

This paper focuses on constructing statistics that average over all of these axes simultaneously, but in practice they have different properties:

1. Computing expectations over axis 1 is typically cheap, since it only requires running the policy for 100 or 1000 episodes.
2. Computing the expected return of an *algorithm* (axis 2) is extremely expensive: each sample requires training a policy to convergence. However, in many practical settings, having even one seed that gives a policy with extremely high expected return is sufficient. Consider AlphaGo: having even one run that produces the best Go player in the world is all you need.
3. Finally, the set of environments is not a stochastic axis, unlike the other two. Every environment is qualitatively different.

This leaves me with questions like “Is comparing algorithms still difficult given a (free) policy evaluation oracle?” and “How much of the difficulty of evaluation comes from aggregating over environments?”

**Scientific testing is not new**

This is a somewhat minor quibble, but the paper frames scientific testing of specific properties of an algorithm as a new proposal. However, I would already consider evaluations of this sort to be standard practice? For example, [Unifying Count-Based Exploration and Intrinsic Motivation](https://arxiv.org/abs/1606.01868) does rely heavily on expected return as an evaluation mesure, but they also investigate the accuracy of the pseudo-count function and the state visitation of the algorithm on Montezuma’s Revenge. [Addressing Function Approximation Error in Actor-Critic Methods](https://arxiv.org/abs/1802.09477) investigates in depth the role of variance and overestimation bias in value function approximation.

**Statistical challenges remain in scientific testing**

The authors propose scientific testing as a solution to the statistical problems of rigorous evaluation of policy return, but I wonder if those same statistical problems do not crop up in scientific testing as well. In a sense the authors have convinced me of too much. In the first half of the paper, they showed that estimating the expected value of a scalar stochastic summary statistic is too expensive to do in RL. But in the second half they propose estimating the expectation of some other scalar summary statistics. Why should I have any confidence in these values?

---

> ### Author Response · Authors · 2023-03-15
> **Reponse to your review**
>
> Thank you for your response. We have made changes to the paper based on your feedback and answered your questions below.
>
>
> >“This is a somewhat minor quibble, but the paper frames scientific testing of specific properties of an algorithm as a new proposal.” and “There needs to be some acknowledgement of the prior existing of scientific testing.”
>
> Yes, scientific testing is not new and there are examples of these types of experiments throughout the history of RL. Although they are not ubiquitous nor the focus of most papers. We have updated the language of the paper to reflect this fact better.
>
>
> >“But in the second half they propose estimating the expectation of some other scalar summary statistics. Why should I have any confidence in these values?”
>
> Yes, statistical challenges will still exist in scientific testing. We have added Section 4.6 to address your concern. The short response is that with scientific testing, we get to specify interesting questions that we are able to remove sufficient statistical uncertainty with the computational resources we have available to us. This is not always easy to do, but scientific testing can answer many more interesting questions than benchmarking. It is the scientist's job to figure out the interesting question that can be answered.
>
> >“This paper focuses on constructing statistics that average over all of these axes simultaneously, but in practice they have different properties”
>
> Yes, examining the aggregate performance combines all of these axes, but this is often the goal in benchmarking algorithm performance.
>
> >“Is comparing algorithms still difficult given a (free) policy evaluation oracle?”
>
> Great question. The answer depends on how much variance is in evaluating the policy versus the variance in the training procedure. If algorithms consistently train to produce a specific policy, then the only remaining variance would be in the policy's performance. In this case, having a policy evaluation oracle would mean you only need to run the algorithm once per environment. However, this no-variance training procedure will only happen if the algorithm converges to an optimal policy. Based on results in the community and our own experience, there is high variance in the training procedure, particularly when considering the need to select hyperparameters. Taking this argument even further, let's say one day we will have many reliable algorithms converging close to the optimal policy. Then each algorithm will have low variance over the training procedure, but they will all have similar levels of performance. Since their performances are similar, it will become harder to tell them apart.
>
>
> >“How much of the difficulty of evaluation comes from aggregating over environments?”
>
> Comparing across multiple environments does add substantial difficulty. If one only wants to compare performance on a single environment, then one can disregard much of the complexity of creating an aggregate performance measure and use standard confidence interval techniques to quantify uncertainty. Often researchers want to make claims about how well their algorithm works across a set of environments. Thus they have to aggregate. However, even in the single environment case, if the performances of two algorithms are similar, it will still require many trials to identify which is better.

---

### Author Response · Authors · 2023-03-15
**Comment to all reviewers**

Dear Reviewers,

Thank you for your feedback on our paper. We have carefully considered your comments and made several revisions to improve the paper. Specifically, we updated the language to reflect that scientific testing is not a new concept in the community. We also conducted a survey of accepted RL papers at NeurIPS 2022 to support our argument that benchmarking is the dominant paradigm of experimentation. We found that 91% of non-theory focused papers included benchmarking experiments, whereas only 35% included one or more scientific testing experiments. Details of the survey can be found in Appendix A.

Regarding concerns raised by several reviewers, we would like to clarify that we are not suggesting that benchmarking experiments should be eliminated entirely. Rather, we propose that scientific testing should be the **primary** form of experimentation. A central assumption in this paper is that the primary experiments in a paper should provide insightful knowledge and have statistically significant results. We provide evidence that benchmarking experiments are often too expensive to produce statistically significant results and that they do not provide insights into how an algorithm works, which limits its ability to serve as a primary form of experimentation. In contrast, scientific testing can offer deeper insights, answer questions that cannot be addressed with benchmarking alone, and have statistically significant results. Thus, we argue that it should be the primary form of experimentation.

We acknowledge that not everyone will agree with our viewpoint, particularly when it comes to reviewing criteria. However, we believe that this paper can contribute to a productive debate within the community about acceptable standards and ways to improve experiments. In this debate, most recent papers emphasize how to do better benchmarking. Instead, we propose that researchers consider other experiments that can offer greater insights. By presenting these arguments, we hope to encourage readers to think critically about what they want to learn from their experiments and to engage in meaningful discussions about the best approaches.

Thank you again for your time and feedback.



Best regards,

The Authors

---

### Decision · Action_Editors · 2023-04-13

**Recommendation:** Reject

**Comment:**

The reviewers agree, and I concur, that the authors make a compelling case for the scientific testing. However, the paper makes overly strong emphasis on using the scientific testing in lieu of the benchmarking (Reviewer znDc: ""scientific testing" paradigm doesn't suffer from the same statistical challenges remains not well supported after their paper update" and "the idea that RL results are very noisy (and thus require lots of trials for confident conclusions) has been pretty well covered in the literature so far. This paper doesn't seem to have much more to say, other than suggesting the community focus on more targeted investigative experiments, which is already common.").

Even after the discussion, the conclusion seems overstated. While the topics is interesting and worth discussing, the declarative nature of the conclusions renders the paper in its current form not suitable for a publication at TMLR. Specifically, the authors argue for replacing benchmarking with scientific testing, which could slow down the progress of the field (Reviewer SL9E: "A paper written with a more balanced viewpoint which address both the limitations of scientific testing and benchmarking would be much more valuable. If everyone in the RL community adopted what the authors are arguing for, it's plausible that no one benchmarks RL algorithms on any problem, which would simply slow down the progress towards solving real-world RL problems. Algorithms like PPO / DQN wouldn't have existed as they stem from non-rigorous benchmarking.").

The authors might want to either consider a different venue, one that allows for more discussion, or revise the future version of the manuscript to make the case and develop recipes for how scientific testing fits in the ablation studies.

**Audience:**

The paper would be of interest on the RL community focused on the traditional RL algorithm developments.

**Claims And Evidence:**

The paper makes a case for scientific testing in lieu of benchmarking for RL algorithm development.  However, reviewers (znDc and SL9E) expressed the concern, and I concur, that the paper makes overly strong emphasis on using the scientific testing in lieu of the benchmarking. Even after the discussion, the conclusion remains declarative in nature (see reviewer znDc comments).

While the topics is interesting and worth discussing, the declarative nature of the conclusions renders the paper in its current form not suitable for a publication at TMLR. Specifically, the authors argue for replacing benchmarking with scientific testing, which could slow down the progress of the field (see reviewer's SL9E comments).

The authors might want to either consider a different venue, one that allows for more discussion, revise the future version of the manuscript to make the case and develop recipes for how scientific testing fits in the ablation studies, or write a survey and future out look on how to benchmark and evaluate RL algorithms.